# Differential Transcriptomic Features of Peripheral Blood Mononuclear Cells in Pulmonary Sarcoidosis with and Without Extrapulmonary Lesions in an East Asian Population

**DOI:** 10.3390/biomedicines13122998

**Published:** 2025-12-07

**Authors:** Yushi Murai, Takeshi Kawasaki, Takuro Imamoto, Daisuke Ishii, Keiichiro Yoshioka, Yoshinori Hasegawa, Osamu Ohara, Koichiro Tatsumi, Takuji Suzuki

**Affiliations:** 1Department of Respirology, Graduate School of Medicine, Chiba University, Chiba 260-8670, Japan; 2Department of Applied Genomics, Kazusa DNA Research Institute, Chiba 292-0818, Japan

**Keywords:** sarcoidosis, extrapulmonary lesions, granuloma formation, peripheral blood mononuclear cells, transcriptomics, gene expression

## Abstract

**Background**: Sarcoidosis is a systemic granulomatous disease of unknown etiology. Pulmonary sarcoidosis with extrapulmonary lesions (EPL) confers poor prognoses. The transcriptomic features of peripheral blood mononuclear cells (PBMCs) could be crucial in sarcoidosis pathogenesis. However, the gene expression characteristics associated with EPL development remain unknown. **Methods**: Bulk PBMCs were collected from 26 healthy controls and 14 patients with pulmonary sarcoidosis stratified into those with (*n* = 9) or without (*n* = 5) EPL. None of the participants were receiving immunosuppressive agents. PBMC transcriptomic analysis was conducted using RNA sequencing. **Results**: Principal component analysis (PCA) revealed a clear distinction between pulmonary sarcoidosis and healthy control groups, with 227 differentially expressed genes (88 upregulated, 139 downregulated), including upregulated (*CLEC7A*, *GBP5*, *JAK2*, *IL15*, *IL1B*, *CXCL8*, and *CXCL10*) and downregulated (*TNFRSF13C*, *CD40LG*, *CD28*, and *ID3*) genes in pulmonary sarcoidosis group. Enrichment analysis revealed upregulated immunological pathways related to granuloma formation in pulmonary sarcoidosis PBMCs, including T helper 17 and tumor necrosis factor-alpha signaling pathways, IL-1B, IL-6, and IL-17 production, and response to external stimuli. Furthermore, patients with and without EPL showed 206 differentially expressed genes (131 upregulated, 75 downregulated), including upregulated (*IFNG* and *IFNLR1*) and downregulated (*SOCS3*, *MMP9*, and *CXCL10*) genes. Gene ontology (GO) analysis revealed that interleukin 6 (IL-6) and IL-23 production were upregulated in patients with EPL. **Conclusions**: These findings elucidate the mechanisms underlying granuloma formation in sarcoidosis and demonstrate the differential transcriptomic features of PBMCs in patients with and without EPL. The upregulation of *IFNG* and *IFNLR1* may be related to EPL development and could serve as potential therapeutic targets for sarcoidosis.

## 1. Introduction

Sarcoidosis is a systemic granulomatous disease of unknown etiology. Its development could be influenced by genetic factors, environmental stimuli, and both foreign and self-antigens [1]. Granuloma formation is initiated by antigen-presenting cells, such as activated monocytes/macrophages, which phagocytose various antigens. These cells subsequently activate T cells, leading to the secretion of inflammatory cytokines, primarily including interferon gamma (IFN-γ) [2]. Mononuclear cells, including monocytes/macrophages and lymphocytes, play a central mechanistic role in granuloma formation in sarcoidosis.

Pulmonary sarcoidosis is the most frequent manifestation of the disease, with pulmonary involvement observed in 96% of a large multicenter cohort including Asian, White, and Black patients [3]. Nevertheless, the mortality rate associated with pulmonary sarcoidosis is lower in Japan than in North America [4,5]. In patients with pulmonary sarcoidosis, extrapulmonary involvement, particularly in cardiac sarcoidosis, is associated with poor prognosis [6]. The pathobiological differences between patients with pulmonary sarcoidosis and concurrent extrapulmonary lesions (EPL) and those without EPL remain unclear.

Peripheral blood mononuclear cells (PBMCs) are immunocompetent cells comprising monocytes and lymphocytes. Monocytes, which are CD14^+^ cells, are further categorized into three distinct populations based on CD14 and CD16 expression levels: CD14^++^CD16^−^ (classical), CD14^++^CD16^+^ (intermediate), and CD14^+^CD16^+^ (non-classical). Lymphocytes are primarily classified into T, B, and natural killer (NK) cells, each exhibiting distinct functions. Since PBMCs comprise diverse cell types, they are considered to play crucial roles in immune-related diseases, including sarcoidosis [7].

Omics approaches, including genomics, epigenomics, transcriptomics, proteomics, and metabolomics, are instrumental for elucidating polygenic and phenotypically diverse diseases. Transcriptomics, in particular, has been applied to investigate sarcoidosis pathogenesis at the PBMC level. RNA sequencing provides comprehensive gene expression profiles, offering detailed insights into disease mechanisms.

A previous study demonstrated differences in PBMC gene expression patterns between patients with sarcoidosis and healthy controls (HCs), though a part of the targeted patients had received steroids and/or immunosuppressant drugs [8]. Our previous report also indicated the differential transcriptomic features of PBMCs between patients with pulmonary sarcoidosis and HCs, and suggested that those features would be reflected by the mechanistic roles of granuloma formation. However, this study had several limitations, notably a statistically significant age disparity between groups and a small sample size [9]. Hence, the transcriptomic analyses of PBMCs in pulmonary sarcoidosis are confounded by factors such as age, sex, or immunosuppressant drug administration. Therefore, in the current study, we analyzed a larger cohort of patients with sarcoidosis and HCs, of similar age and sex distribution, to evaluate how PBMC gene expression is associated with sarcoidosis lesion progression. In addition, we investigated gene expression patterns associated with the development of EPL in pulmonary sarcoidosis.

## 2. Materials and Methods

### 2.1. Subjects

This study was approved by the Human Ethics Committee of Chiba University (protocol number: 2083). PBMCs were collected from patients diagnosed with pulmonary sarcoidosis and from HCs between August 2020 and November 2021 at Chiba University Hospital. The pulmonary sarcoidosis diagnosis in the patients was based on the recommendations of the American Thoracic Society (ATS), European Respiratory Society (ERS), and World Association of Sarcoidosis and Other Granulomatous Diseases (WASOG) statement [10]. Patients lacking histopathological confirmation of granulomas were excluded from the present study. This study included 14 patients with pulmonary sarcoidosis and 26 HCs. The demographic characteristics of the participants are presented in Table 1. Eight patients had a history of smoking, and none were receiving any immunosuppressive agents. Nine patients also presented with extrapulmonary sarcoidosis. All patients with pulmonary sarcoidosis exhibited lung involvement, which was classified using chest radiographic stages 1, 2, or 3. No significant differences in age or sex were observed between the pulmonary sarcoidosis and HC groups.

The demographic characteristics of patients with and without EPL are presented in Table 2. No significant differences in age or sex were observed between the two subgroups. We defined an extrapulmonary lesion as active organ involvement of sarcoidosis other than the lung at the time of PBMC sample collection. All patients were evaluated for cardiac and ocular involvement by cardiologists and ophthalmologists, respectively. Cardiac sarcoidosis was diagnosed according to the Japanese Circulation Society (JCS) criteria [11], and ocular sarcoidosis was diagnosed using the International Workshop on Ocular Sarcoidosis (IWOS) criteria [12]. Potential involvement of other organs was assessed by appropriate specialists when patients presented with relevant symptoms. The details of the EPL are provided in Table 3. Patients were followed for a median duration of 38 months (range, 1–57 months). None of the patients with pulmonary sarcoidosis without EPL developed EPL during the follow-up period.

### 2.2. Isolation of PBMCs

Peripheral blood was collected using BD Vacutainer CPT Cell Preparation Tubes with sodium citrate according to the manufacturer’s protocol (Cat#362760; Becton, Dickinson and Company, Franklin Lakes, NJ, USA). Briefly, blood-containing tubes were centrifuged at 1500 rpm for 20 min at room temperature. After centrifugation, the plasma was removed from the uppermost layer. The PBMC layer was then transferred to 15 mL conical tubes. The PBMC were washed twice with phosphate-buffered saline (PBS), resuspended in Isogen (Nippongene, Tokyo, Japan), and stored at −80 °C.

### 2.3. Total RNA Extraction, mRNA Library Preparation, and 3′ RNA-Seq

Total RNA was extracted from 1.0–2.0 × 10^6^ PBMCs dissolved in 1.0 mL of Isogen reagent. The solution was vigorously vortexed and incubated at room temperature for 5 min. Chloroform was then added, and the mixture was centrifuged. The aqueous phase was carefully transferred to a new tube, followed by the addition of 10 μg of glycogen (Life Technologies, Carlsbad, CA, USA) as a co-precipitant. RNA was precipitated using 600 µL of isopropyl alcohol, washed with 75% ethanol, and dissolved in 10 µL of RNase-free water. The concentration and quality of the RNA were verified using a Qubit fluorometer (Life Technologies) and an Agilent 2100 Bioanalyzer (Agilent Technologies, Santa Clara, CA, USA), respectively. Purified total RNA (200 ng) was used for RNA library preparation, following the Quant-Seq 3′ mRNA-seq library preparation kit FWD protocol for Illumina (Lexogen, Vienna, Austria). The RNA libraries were sequenced on an Illumina NextSeq 500 system (Illumina Inc, San Diego, CA, USA) with 75-nucleotide-long reads.

### 2.4. 3′ RNA-Seq Data Analysis

RNA-seq reads per million (RPM) data were analyzed using the Qlucore Omics Explorer software version 3.9.9 (Qlucore AB, Lund, Sweden). Differentially expressed genes (DEGs) were identified based on a log_2_-fold change threshold of >2.0 (upregulated) or <0.5 (downregulated) with *p*-values of <0.01 for comparisons between patients with pulmonary sarcoidosis and HCs. For comparisons between pulmonary sarcoidosis with and without extrapulmonary lesions, the same log_2_-fold change criteria were applied, with *p*-values of <0.05. The false discovery rate (FDR) was also calculated as a *q*-value and taken into consideration to interpret the *p*-values. Gene Ontology (GO) and KEGG pathway enrichment analyses were conducted using the Enrichr online tool (http://amp.pharm.mssm.edu/Enrichr/ accessed on 7 December 2023) to evaluate the enrichment analysis of DEGs. The gene set databases used were “GO_Biological_Process_2023” (Terms: 42,312; gene coverage: 1,536,078) and “KEGG_2021_Human” (Terms: 320; gene coverage: 8078).

### 2.5. Statistical Analysis

Age characteristics of the samples are expressed as means ± standard deviation (SD).

Levene’s test was used to evaluate equal variance between groups. Student’s *t*-test was used for age comparisons, whereas Fisher’s exact test was applied to compare gender distributions. Statistical significance was defined at *p* < 0.05. We performed statistical power analyses to estimate our ability to detect DEGs in each comparison using RNASeqPower package (Bioconductor version 3.21). Input parameters included the observed sequencing depth (average reads per gene), a biological coefficient of variation (CV) of 0.4, the nominal significance level (α) set according to the *p*-value threshold for each analysis, and the effect size defined as a log_2_-fold change of >2.0 or <0.5.

## 3. Results

### 3.1. Differential Gene Expression and Pathway Analysis in PBMCs Between Patients with Pulmonary Sarcoidosis and Healthy Controls

RNA sequencing libraries were prepared from mRNA isolated from PBMCs of the study participants: 14 patients with pulmonary sarcoidosis and 26 HCs. The RNA integrity numbers of all samples were >8, and 26,469 genes were initially detected. After applying additional quality control measures, 12,331 genes were retained for subsequent analysis. The estimated power to detect differentially expressed genes (DEGs) was approximately 0.99 for 14 patients with sarcoidosis vs. 26 HCs, at a sequencing depth of 49, and α = 0.01, based on a log_2_-fold change of >2.0 or <0.5. This power was considered sufficient to obtain reliable results.

Principal component analysis (PCA) demonstrated a clear distinction between the two groups (Figure 1A). Subsequently, we compared DEGs in the PBMCs of patients with pulmonary sarcoidosis and HCs. The distribution of the log_2_-fold changes and the *p*-values for the 12,331 expressed genes in these samples are presented in a volcano plot, with 227 DEGs highlighted in color (*p* < 0.01, fold change >2 or <0.5). Notable upregulated genes such as *CLEC7A*, *GBP5*, *JAK2*, *IL15*, *IL1B*, *CXCL8*, and *CXCL10* and downregulated genes such as *TNFRSF13C*, *CD40LG*, *CD28*, and *ID3* were labeled (Figure 1B). A heatmap of the 227 DEGs, comprising 88 upregulated and 139 downregulated genes in the PBMCs, revealed distinct transcriptomic signatures between the two groups (Figure 1C). All DEGs were listed in Appendix A.

Enrichment analysis identified various biological processes and pathways associated with the DEGs between patients with pulmonary sarcoidosis and HCs. Notably, the enriched biological process terms included positive regulation of interleukin (IL)-1 beta production, phagocytosis, T-helper (Th) 17-type immune response, and type II interferon production. The enriched pathway terms included tumor necrosis factor (TNF), IL-17, and Toll-like receptor signaling pathways, indicating the activation of immune responses promoting granuloma formation. Gene Ontology (GO) terms for upregulated and downregulated genes are listed in Table 4, whereas Kyoto Encyclopedia of Genes and Genomes (KEGG) pathway terms are provided in Table 5. All data of enrichment analysis were listed in Appendix A.

### 3.2. Differential Gene Expression and Pathway Analysis in PBMCs Between Patients with Pulmonary Sarcoidosis with and Without EPL

To investigate potential prognostic factors in pulmonary sarcoidosis, we stratified patients with pulmonary sarcoidosis into two groups: those with (*n* = 9) and without (*n* = 5) EPL. After processing the RNA sequencing libraries, 12,453 genes were identified in patients with pulmonary sarcoidosis, with and without EPL. The estimated power to detect DEGs was approximately 0.84 with 9 with EPL vs. 5 without EPL, at a sequencing depth of 61, and α = 0.05, based on a log_2_-fold change of >2.0 or <0.5. Although the sample size was smaller than that of the comparison between patients with sarcoidosis and HCs, the power was still acceptable, and the results are likely to be reliable.

To investigate potential prognostic factors in pulmonary sarcoidosis, we stratified patients with pulmonary sarcoidosis into two groups: those with (*n* = 9) and without (*n* = 5) EPL. After processing the RNA sequencing libraries, 12,453 genes were identified in patients with pulmonary sarcoidosis, with and without EPL. The estimated power to detect DEGs was approximately 0.84 with 9 with EPL vs. 5 without EPL, at a sequencing depth of 61, and α = 0.05, based on a log_2_-fold change of >2.0 or <0.5. Although the sample size was smaller than that of the comparison between patients with sarcoidosis and HCs, the power was still acceptable, and the results are likely to be reliable.

Principal component analysis demonstrated distinctions between the two groups (Figure 2A). The distribution of the log_2_-fold changes and the *p*-value for the 12,453 genes is depicted in a volcano plot, with 206 DEGs (*p* < 0.05, fold change >2 or <0.5) highlighted in color. Notable upregulated genes, such as *IFNG* and *IFNLR1*, and downregulated genes, such as *MMP9*, *SOCS3*, and *CXCL10*, were labeled. (Figure 2B). A heatmap of the 206 DEGs, comprising 131 upregulated and 75 downregulated genes in the PBMC, revealed distinct transcriptomic signatures between the two subgroups (Figure 2C). All DEGs were listed in Appendix A.

Enrichment analysis indicated that various biological processes and pathways were associated with the DEGs between the two subgroups. Notably, the enriched upregulated biological process terms included the positive regulation of cellular respiration and interferon-mediated signaling pathway, which included interferon lambda receptor 1 (*IFNLR1*) and *IFNG* as DEGs. The enriched downregulated pathways included the TNF and IL-17 signaling pathways, which included matrix metalloproteinase-9 (*MMP9*), C-X-C motif chemokine ligand 10 (*CXCL10*) and suppressor of cytokine signaling 3 (*SOCS3*) as DEGs. The GO terms for the upregulated and downregulated genes are listed in Table 6. The KEGG pathway terms are provided in Table 7. All data of enrichment analysis were listed in Appendix A.

Furthermore, an exploratory comparison was conducted between two patients with cardiac involvement in pulmonary sarcoidosis and patients without EPL. The estimated power to detect DEGs was approximately 0.48 for 2 patients with cardiac sarcoidosis vs. 5 without EPL, at a sequencing depth of 39, and α = 0.05, based on a log_2_-fold change of >2.0, or <0.5. Because of the limited power, these results should be considered exploratory rather than conclusive.

After processing the RNA sequencing libraries, 17,153 genes were identified in patients with pulmonary sarcoidosis with and without cardiac involvement. Principal component analysis revealed a separation between the two groups (Appendix A). The distribution of the log_2_-fold changes and the *p*-value for the 17,153 genes is depicted in a volcano plot, with 449 DEGs (*p* < 0.05, fold change > 2 or <0.5) highlighted in color (Appendix A). A heatmap of the 449 DEGs, comprising 259 upregulated and 190 downregulated genes in PBMCs, revealed distinct transcriptomic signatures between the two groups (Appendix A).

Enrichment analysis indicated that various biological processes and pathways were associated with the DEGs between these two groups, including nitric oxide synthase (*NOS*) and adenylate cyclase (*ADCY*). The GO terms for the upregulated and downregulated genes are listed in Appendix A. The KEGG pathway terms are provided in Appendix A. This subgroup analysis was exploratory because it was based on an extremely small sample of patients with cardiac sarcoidosis (*n* = 2); therefore, the results should be interpreted with caution.

## 4. Discussion

In this study, transcriptomic analysis revealed that distinct gene expression profiles differed in the PBMCs of patients with pulmonary sarcoidosis compared to those of HCs. Enrichment analyses indicated the upregulation of various terms and pathways associated with granuloma formation in sarcoidosis. These findings provide a more detailed understanding of the pathobiological mechanisms underlying granuloma formation in pulmonary sarcoidosis. Furthermore, this study demonstrates the differences in the transcriptome features of PBMCs in pulmonary sarcoidosis between patients with and without EPL, suggesting that gene expressions in PBMCs may reflect the clinical features of sarcoidosis involving multiple organs.

### 4.1. Patients with Pulmonary Sarcoidosis vs. Healthy Controls (HCs)

Transcriptomic analysis of bulk PBMCs identified 227 differentially expressed genes between patients with pulmonary sarcoidosis and HCs. GO and KEGG pathway enrichment analyses highlighted numerous immunological responses, inflammatory cytokine production pathways, and responses to external stimuli (Table 4 and Table 5). These results supported well-established pathobiological mechanisms underlying sarcoidosis. Our previous study indicated the differential transcriptomic features of PBMCs in pulmonary sarcoidosis could be associated with the mechanism of sarcoidosis; however, this study had several limitations, notably a statistically significant age disparity between groups and a small sample size [9]. Therefore, we used a larger cohort where no significant differences were observed in age and sex between the pulmonary sarcoidosis and HCs to verify the findings in our previous study. Although still exploratory, the age/sex matching and the larger sample size in the current study provide more confidence in the observed gene expression signatures.

First, our analysis revealed that pathways related to responses to external stimuli were upregulated in patients with pulmonary sarcoidosis. GO terms such as “response to molecules of bacterial origin”, “positive regulation of response to external stimuli,” and “response to lipopolysaccharide,” were enriched. Lipopolysaccharide, a component of the outer membrane of Gram-negative bacteria, triggers innate immune responses primarily through Toll-like receptor 4 (TLR-4), leading to the production of inflammatory cytokines such as TNF-α, IL-1, IL-6, and IL-8 in monocytes and macrophages—key mediators involved in granuloma formation in sarcoidosis [13].

Furthermore, the “Toll-like receptor signaling pathway” was identified in the KEGG pathway analysis. TLR-2 and TLR-4 expression in blood mononuclear cells is significantly higher in patients with sarcoidosis than that in HCs [14]. Similarly, the “nucleotide oligomerization domain (NOD)-like receptor signaling pathway” was enriched, highlighting the role of NOD-like receptors (NLRs) in activating transcriptional factors such as nuclear factor kappa B (NF-κB), activator protein-1 (AP-1) and interferon regulator factor 5 (IRF5) that drive pro-inflammatory cytokine production [15]. NLRs comprise cytosolic proteins known as nucleotide-binding oligomerization domains (NODs), including NOD1 and NOD2. Notably, hyperfunction mutations in NOD2 cause Blau syndrome and early-onset sarcoidosis, which are systemic granulomatous diseases [16]. Additionally, the “C-type lectin receptor signaling pathway” was identified with the C-type lectin domain family 7 member A (*CLEC7A*). *CLEC7A* is expressed in monocytes, macrophages, neutrophils and dendritic cells, and is involved in antifungal immunity [17]. *CLEC7A* activation induces cytokine release, such as IL-1β, IL-6 and IL-23, favoring Th17 cell differentiation, and IL-12, driving IFN-γ production by Th1 and NKT cells [18].

Gene Ontology and KEGG pathway enrichment analyses also highlighted terms related to phagocytosis; specifically, “positive regulation of phagocytosis” and “phagosome” were listed. Phagocytosis is a cellular process that engulfs and clears microbes and apoptotic cells, producing inflammatory cytokines, and antigen presentation, particularly by macrophages, monocytes and dendritic cells. These cells initiate the differentiation of Th1 and 17 cells or produce IFN-γ, potentially contributing to granuloma formation in pulmonary sarcoidosis [19]. Collectively, these results suggest that pulmonary sarcoidosis involves an immunological response to unidentified external antigens.

Several pathways involved in cytokine regulation were also enriched, including “positive regulation of IL-1β production”, “positive regulation of IL-6 production,” and “TNF-α signaling pathway”. These cytokines are produced by alveolar macrophages in sarcoidosis even in unstimulated conditions [20]. The elevated expression of these cytokines in PBMCs suggests their potential role in the systemic granulomatous mechanisms of sarcoidosis.

Pathways related to Th17 were also enriched, including “positive regulation of Th17 type immune response,” “positive regulation of IL-17 production,” “IL-17 signaling pathway,” and “Th17 cell differentiation.” Th17 cells, which differentiate from CD4^+^ T cells under IL-6 and transforming growth factor beta (TGF-β) stimulation, are implicated in granuloma formation [21]. However, Th17.1 cells, known for producing high amounts of IFN-γ, are considered more crucial for maintaining granuloma [22]. Interestingly, Th17.1 cells are enriched in sarcoidosis lesions but are less prevalent in the peripheral blood [23], possibly explaining why terms related to Th17.1 cells were not enriched in PBMCs in this study.

Finally, the “cellular response to type II interferons” and the “positive regulation of type II interferon production” were identified. IFN-γ, produced by natural killer (NK) cells, Th1 cells, Th17.1 cells and CD8^+^ T cells, activates macrophages, promoting TNF-α and IL-12 production by macrophages [24]. Macrophages produce chemokine ligands such as monocyte chemoattractant protein-1 (MCP-1), C-C motif chemokine ligand 20 (CCL20), CXCL10 and CXCL16 under stimulation of both TNF-α and IFN-γ, thereby attracting Th1/17 cells, monocytes, regulatory T and B cells to inflammation sites [19]. The upregulation of *CXCL10*, associated with “regulation of monocyte chemotaxis”, aligns with previous findings on the roles of IFN-γ in granuloma formation [1].

### 4.2. Pulmonary Sarcoidosis with vs. Without Extrapulmonary Lesions (EPL)

The current results revealed differential transcriptomic features of PBMCs in pulmonary sarcoidosis between patients with and without EPL, although our previous study found no transcriptomic differences between the two groups, probably due to a small sample size [9]. The current analysis successfully identified 206 DEGs associated with EPL. This finding identified a novel *INFG*/*INFLR1* signature. Therefore, this study provides the first transcriptomic evidence differentiating systemic EPL sarcoidosis from isolated pulmonary disease. The present study has shown the upregulation of *IFNG* and *IFNLR1*, included in the “interferon-mediated signaling pathway,” in patients with sarcoidosis with EPL manifestation, suggesting their potential role in granuloma progression to multiple organs. Interferon gamma (*IFNG*) is the gene that encodes IFN-γ, a member of the type II interferon family and a key cytokine in sarcoidosis pathogenesis. Single-cell analyses of granulomas in skin sarcoidosis have revealed that *IFNG* is upregulated in helper T cells, including Th17.1 cells, but not in macrophages [25]. Th17.1 cells are more prevalent in the bronchoalveolar lavage fluid than in the peripheral blood from patients with pulmonary sarcoidosis [23,26]. Our study suggests that the upregulation of *IFNG* in the PBMCs may closely reflect the activities of systemic granuloma formation and contribute to EPL during sarcoidosis.

*IFNLR1,* which belongs to the class II cytokine receptor family, forms a receptor complex with IL-10 receptor 2 (IL10R2). Its ligands are known as IL-28A, IL-28B, and IL-29, categorized as type III interferons (IFN-λ) [27]. The *IFNLR1* expression has been detected in various immune cells, including macrophages, dendritic cells, neutrophils, and B and T lymphocytes, whereas minimal to no expression has been observed in monocytes and NK cells [28,29]. IFN-λ is secreted by mucosal epithelial cells in response to viral infection and regulates inflammatory responses [30]. The IFN-λ signaling pathway plays a pivotal role in guiding the host response to numerous pathogens encountered at mucosal surfaces [31]. Furthermore, a recent study on macaque monkeys infected with *Mycobacterium tuberculosis* revealed that IFN-λ and IFNRL1 are expressed in lung granulomas, and IFN-λ signaling is partly driven by TLR2 ligation [32]. These findings indicate that IFN-λ may be closely associated with granuloma formation and maintenance.

*IFNG* and *IFNRL1* are included in several GO terms: “positive regulation of IL-23 production,” “regulation of IL-6 production,” and “interferon-mediated signal pathway”. IL-6 induces Th1 and Th17 cell differentiation, and the survival and proliferation of Th17 cells depend on IL-23. These cytokines play important roles in granuloma development by activating Th1 and Th17 cells.

In sarcoidosis, the *IFNLR1* functions and regulatory mechanisms are yet to be fully elucidated. However, our study suggests that *IFNLR1* upregulation in the PBMCs of patients with pulmonary sarcoidosis with EPL may relate to EPL development through PBMC recruitment to local lesions via circulation. IFN-λ may promote Th1 polarization in T cells, thereby enhancing macrophage activation and contributing to granuloma formation [32]. Taken together, IFNLR1 upregulation in the PBMCs suggests that IFN-λ signaling could drive granuloma formation not only in the lung but also in multiple organs.

Meanwhile, the “TNF signaling pathway” and “IL-17 signaling pathway” were listed in the KEGG pathway analysis with downregulated genes, including *MMP9*, *CXCL10*, and *SOCS3*. *SOCS3* is a negative regulator of the Janus kinase/signal transducers and activators of the transcription (JAK/STAT) pathway, especially in STAT3, which activates the IL-23 signaling pathway. In contrast, the loss of *SOCS3* leads to enhanced Th17 generation [33]. Furthermore, Th17.1 cells, which produce both IL-17 and IFN-γ, display reduced *SOCS3* expression compared to that of Th17 or Th1 cells [34]. Therefore, *SOCS3* downregulation may enhance Th17.1 polarization.

*MMP9* degrades the extracellular matrix and promotes fibrosis [25]. Its expression is stronger in later-stage granulomas with less lymphocyte infiltration, suggesting that MMP-9 promotes granulomatous fibrosis in the chronic sarcoidosis stage [35]. Furthermore, a lower number of collagen fibers and a lower density of elastic fibers are observed in extrapulmonary granulomas compared with those in pulmonary granulomas in sarcoidosis [36]. Therefore, the downregulation of *MMP9* in the PBMCs from patients with pulmonary sarcoidosis with EPL might indicate the different pathobiological features between extrapulmonary and pulmonary sarcoidosis.

Cardiac sarcoidosis is a clinically poor prognostic factor for patients with pulmonary sarcoidosis. Therefore, we additionally compared the transcriptomes of PBMCs between patients with and without cardiac involvement; however, these results should be considered exploratory rather than conclusive, due to the limited statistical power of our analysis. The results indicated that genes related to cardiac muscle function, including myomesin 2 (*MYOM2*), adenylate cyclase type 2 (*ADCY2*), *ADCY5*, and NOS (including *NOS1* and *NOS2)*, might be upregulated in PBMCs from patients with cardiac involvement. This upregulation led to the enrichment of several GO terms and KEGG pathways.

Regarding the upregulation of *NOS2* in PBMCs, NOS2 is recognized as a marker of M1 macrophages, which exhibit pro-inflammatory functions induced by LPS or IFN-γ and also generate NO from L-arginine to prevent bacterial growth in inflammation sites including granulomas [37]. Although the mechanisms linking NOS2 and cardiac involvement remain unclear, the upregulation of NOS2 in sarcoidosis may be related to the promotion of granuloma formation in inflammatory states. These results were obtained from an exploratory data analysis using a limited sample size; validation studies with larger sample sizes are essential.

This study has several limitations. First, it remains unclear whether the observed changes in PBMC gene expression are a consequence of the disease or if they actively contribute to the disease development. Second, although the DEGs identified in this study could provide insights into the pathological mechanisms of pulmonary sarcoidosis with EPL, these findings have not been confirmed by assessing the corresponding gene expression or protein levels in PBMCs using qPCR or cytokine quantification, nor have they been visualized by using some pathway analysis tools. Additionally, the false discovery rate of the q-value should be taken into consideration to interpret the *p*-values. Furthermore, if cell sorting has been performed prior to the analysis or single-cell RNA sequencing has been carried out, more specific DEGs for each cell type would likely have been detected. Validation of the transcriptomic data would enhance the robustness of the conclusions of this study. Third, this study was conducted at a single center in Japan with a small, ethnically homogenous cohort (although we matched age and sex between groups). Because ethnic differences in sarcoidosis phenotypes and genetic factors have been reported [38], transcriptomic signatures of sarcoidosis may also vary across ethnic groups. Therefore, the generalizability of our results to other populations is limited, and validation studies in more diverse cohorts are warranted. Finally, transcriptomic features of PBMCs from the EPL group may be heterogeneous due to the diversity of extrapulmonary lesions. It is highly likely that the observed gene expression patterns are not uniform across all organ manifestations. Accordingly, studies with a larger stratified cohort would provide clearer organ-specific gene expression profiles. Furthermore, although five patients were classified as without EPL, it remains possible that they had subclinical extrapulmonary involvement that could not be detected even by specialist evaluation.

In conclusion, our study demonstrates that the gene expression profiles in PBMCs differ between patients with pulmonary sarcoidosis and age and sex-matched HCs. Various GO terms and KEGG pathways associated with granuloma formation were identified, strengthening our understanding of the underlying mechanisms in patients with sarcoidosis. Furthermore, the differential transcriptome features in PBMCs from patients with pulmonary sarcoidosis between patients with and without EPL were demonstrated, particularly the upregulations of *IFNG* and *IFNLR1*. The upregulation of these genes might be related to EPL development and could serve as potential therapeutic targets for sarcoidosis.

## Figures and Tables

**Figure 1 biomedicines-13-02998-f001:**
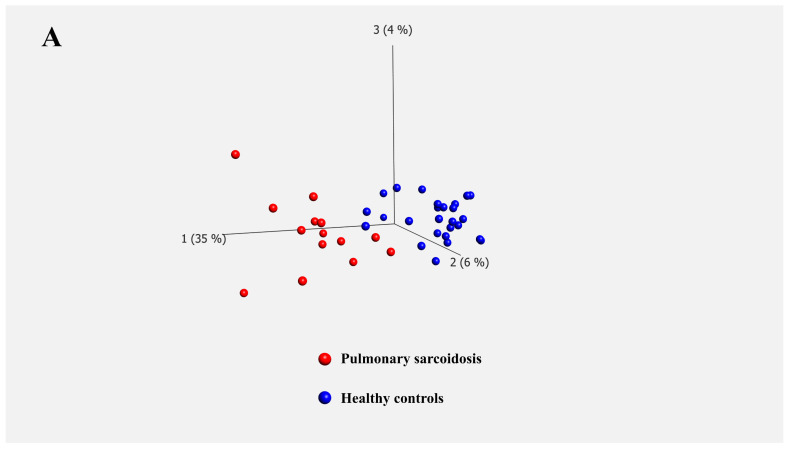
Transcriptomic profiles of peripheral blood mononuclear cells (PBMCs) from patients with sarcoidosis vs. healthy controls. (**A**) Principal component analysis (PCA) separates two well-differentiated groups: patients with pulmonary sarcoidosis and healthy controls (HCs). (**B**) Volcano plot of differentially expressed genes (DEGs) illustrates the distribution of log_2_-fold changes and *p*-values for the 12,331 genes. Colored dots represent the 227 DEGs identified between patients with pulmonary sarcoidosis and HCs. Red dots represent genes with significantly higher expression, whereas blue dots represent genes with significantly lower expression in patients with pulmonary sarcoidosis compared to those in HCs. *CLEC7A*: C-type lectin domain containing 7A, *CXCL8*: C-X-C motif chemokine ligand 8, *CXCL10*: C-X-C motif chemokine ligand 10, *GBP5*: guanylate binding protein 5, *IL1b*: interleukin 1 beta, *IL15*: interleukin 15, *JAK2*: Janus kinase 2, *CD28*: CD28 molecule, *CD40LG*: CD40 ligand, *ID3*: inhibitor of DNA binding 3, *NOG*: noggin, *TNFRSF13C*: tumor necrosis factor receptor superfamily member 13C. (**C**) Heatmap of DEGs illustrates distinct transcriptome signatures of the 227 DEGs between patients with pulmonary sarcoidosis and HCs. Red bars represent high gene expression, and blue bars represent low gene expression.

**Figure 2 biomedicines-13-02998-f002:**
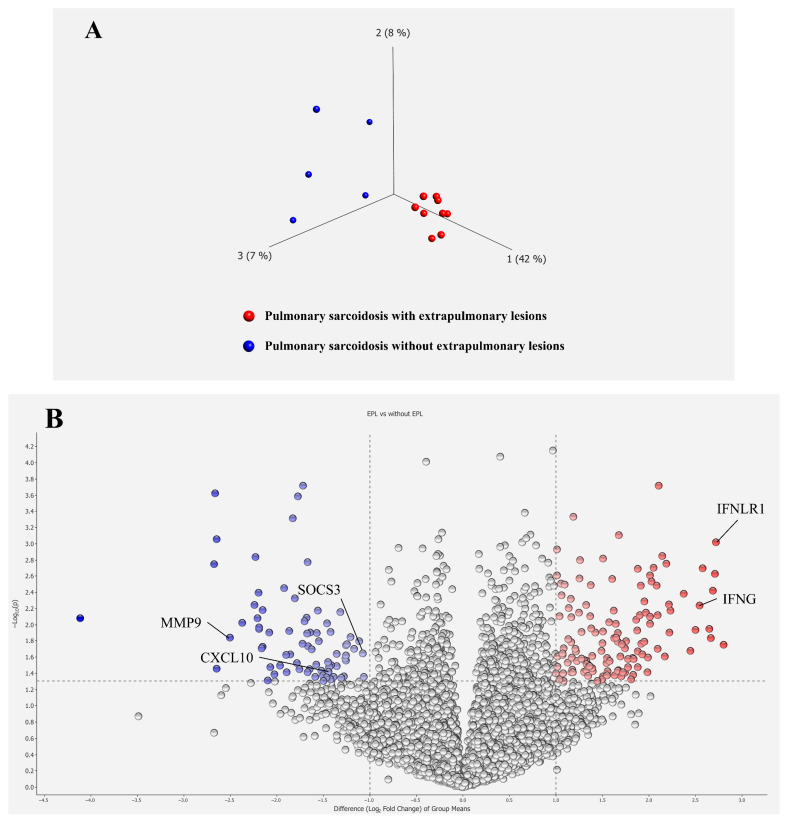
Transcriptomic differences in PBMCs from patients with pulmonary sarcoidosis and without extrapulmonary lesions. (**A**) PCA separates two well-differentiated groups of patients with pulmonary sarcoidosis: those with extrapulmonary lesions (EPL) and those without. (**B**) A volcano plot of DEGs illustrates the distribution of the log_2_-fold changes and *p*-values for the 12,453 genes. Colored dots represent 206 DEGs between patients with pulmonary sarcoidosis with and without EPL. Red dots represent genes with significantly higher expression, whereas blue dots represent genes with significantly lower expression in patients with pulmonary sarcoidosis with EPL compared to those without. *IFNG*: interferon gamma, *IFNLR1*: interferon lambda receptor 1, *CXCL10*: C-X-C motif chemokine ligand 10, *MMP9*: matrix metallopeptidase 9, *SOCS3*: suppressor of cytokine signaling 3. (**C**) Heatmap of DEGs illustrates distinct transcriptome signatures of the 206 DEGs between patients with pulmonary sarcoidosis with and without EPL. Red bars represent high gene expression, and blue bars represent low gene expression.

**Table 1 biomedicines-13-02998-t001:** Characteristics of patients with pulmonary sarcoidosis and health controls.

	Patients with Pulmonary Sarcoidosis	Healthy Controls	*p*-Value
Number of participants	14	26	N/A
Age (mean ± SD)	53.6 ± 15.0	58.1 ± 14.5	0.38
Sex (male/female)	6/8	9/17	0.73

N/A, not applicable; SD, standard deviation.

**Table 2 biomedicines-13-02998-t002:** Characteristics of patients with pulmonary sarcoidosis with and without extrapulmonary lesions (EPL).

	EPL	Without EPL	*p*-Value
Number of participants	9	5	N/A
Age	54.2 ± 18.9	52.6 ± 8.5	0.83
Sex (male/female)	4/5	2/3	1
ACE	30.1	27.8	0.61
CD4/CD8	5.3	7.8	0.38
Stage (≧II/I)	4/5	3/2	1
Mean follow-up period (months)	37	48	0.93

**Table 3 biomedicines-13-02998-t003:** Details of extrapulmonary lesions in patients with pulmonary sarcoidosis follow-up time after sample collection.

Samples	Extrapulmonary Lesions	Follow-Up Period (Month)
No. 1	Eye	37
No. 2	Heart	29
No. 3	Heart, liver, and kidney	1
No. 4	Eye	57
No. 5	Skin	57
No. 6	Eye	48
No. 7	Eye	1
No. 8	Nerve	52
No. 9	Nerve	40
No. 10	Without EPL	39
No. 11	Without EPL	10
No. 12	Without EPL	17
No. 13	Without EPL	2
No. 14	Without EPL	45

**Table 4 biomedicines-13-02998-t004:** Gene Ontology (GO) terms for upregulated and downregulated genes in PBMCs from patients with pulmonary sarcoidosis vs. healthy controls. Relevant terms were excerpted.

Term (GO: Biological Process) with Upregulated Genes	*p*-Value	*q*-Value	DEGs
Positive regulation of interleukin-1 beta production (GO: 0032731)	5.141 × 10^−6^	0.00182	*CLEC7A*, *GBP5*, *JAK2*, *EGR1*, *P2RX7*
Positive regulation of phagocytosis (GO: 0050766)	0.00014	0.00014	*IL15*, *IL1B*, *CLEC7A*
Regulation of monocyte chemotaxis (GO: 0090025)	0.00018	0.00018	*CXCL10*, *SLAMF8*, *DUSP1*
Cellular response to type II interferon (GO: 0071346)	0.00020	0.00990	*GBP5*, *STAT1*, *JAK2*, *GBP1*
Positive regulation of T-helper 17 type immune response (GO: 2000318)	0.00145	0.03076	*CLEC7A*, *JAK2*
Positive regulation of Type II interferon production (GO: 00327729)	0.00280	0.03716	*GBP5*, *CLEC7A*
Response to molecules of bacterial origin (GO: 0002237)	0.00350	0.04828	*CXCL8*, *IL1B*, *JAK2*
Positive regulation of interleukin-6 production (GO: 0032755)	0.00459	0.05879	*SIGLEC16*, *CLEC7A*, *ILB1*
Positive regulation of response to external stimuli (GO: 0032103)	0.00489	0.05992	*GBP5*, *SIGLEC16*, *IL15*, *ILB1*
Positive regulation of interleukin-17 production (GO: 0032740)	0.00496	0.05992	*IL15*, *JAK*
Response to lipopolysaccharide (GO:0032496)	0.00535	0.06145	*CXCL10*, *CXCL8*, *ILB1*, *JAK2*
Positive regulation of interleukin-2 production (GO: 0032743)	0.00538	0.06145	*CLEC7A*, *ILB1*
Term (GO: biological process) with downregulated genes	*p*-value	*q*-value	DEGs
Positive regulation of T cell activation (GO: 0050870)	0.00091	0.3187	*TNFRSF13C*, *CD28*, *CD40LG*, *RHOH*, *CCR7*
B cell-mediated immunity (GO: 0019724)	0.00475	0.3187	*CD27*, *CD19*
B cell activation involved in immune response (GO: 0002312)	0.00540	0.3187	*CD40LG*, *CD19*
Regulation of interleukin-4 production (GO: 0032673)	0.01103	0.3187	*CD40LG*, *LEF1*
Regulation of interleukin-10 production (GO: 0032733)	0.01837	0.3187	*CD40LG*, *CD28*

**Table 5 biomedicines-13-02998-t005:** Kyoto Encyclopedia of Genes and Genomes (KEGG) pathway terms for upregulated and downregulated genes in PBMCs from patients with pulmonary sarcoidosis vs. healthy controls. Relevant terms were excerpted.

Term (KEGG Pathway) with Upregulated Genes	*p*-Value	*q*-Value	DEGs
TNF signaling pathway	3.327 × 10^−8^	3.211 × 10^−6^	*MAP3K8*, *CXCL10*, *SOCS3*, *IL15*, *JAG1*, *PTGS2*, *IL1B*, *FOS*
IL-17 signaling pathway	3.645 × 10^−6^	0.00014	*CXCL10*, *PTGS2*, *CXCL8*, *IL1B*, *FOS*, *FOSB*
Toll-like receptor signaling pathway	6.559 × 10^−6^	0.00020	*MAP3K8*, *CXCL10*, *CXCL8*, *IL1B*, *FOS*, *STAT1*
NOD-like receptor signaling pathway	1.516 × 10^−5^	0.00032	*CXCL8*, *IL1B*, *STAT1*
C-type lectin receptor signaling pathway	9.601 × 10^−5^	0.00120	*CLEC7A*, *PTGS2*, *IL1B*, *STAT1*
Phagosome	0.00056	0.00467	*FCAR*, *FCGR1A*, *MRC1*, *CLEC7A*
Th17 cell differentiation	0.00128	0.00834	*STAT1*, *IL1B*, *FOS*, *JAK2*
JAK-STAT signaling pathway	0.00571	0.02854	*SOCS3*, *IL15*, *STAT1*, *JAK2*
NF-kappa B signaling pathway	0.01089	0.04647	*CXCL8*, *IL1B*, *PTGS2*
Term (KEGG pathway) with downregulated genes	*p*-value	*q*-value	DEGs
TGF-β signaling pathway	0.02784	0.426	*ID3*, *ACVR2B*, *NOG*

**Table 6 biomedicines-13-02998-t006:** Gene Ontology terms for upregulated and downregulated genes in PBMCs from patients with pulmonary sarcoidosis with vs. those without extrapulmonary lesions. Relevant terms were excerpted.

Term (GO: Biological Process) with Upregulated Genes	*p*-Value	*q*-Value	DEGs
Cellular response to pH (GO: 0071467)	0.00029	0.2003	*SCNN1D*, *PKD1L3*, *GPR68*
Mitochondrial protein processing (GO: 0034982)	0.00185	0.2748	*IMMP2L*, *IMMP1L*
Positive regulation of cellular respiration (GO: 1901857)	0.00185	0.2748	*IFNLR1*, *IFNG*
Positive regulation of leukocyte degranulation (GO: 0043302)	0.00317	0.2748	*SPHK2*, *KLRC2*
Positive regulation of signal receptor activity (GO: 2000273)	0.0232	0.3282	*IFNG*, *HFE*
Positive regulation of CD4-positive, CD25-positive, alpha-beta regulatory T cell differentiation (GO: 0032831)	0.03233	0.3282	*IFNG*
Regulation of vitamin D biosynthetic process (GO: 0060556)	0.03233	0.3282	*IFNG*
Positive regulation of interleukin-23 production (GO: 0032747)	0.03867	0.3282	*IFNG*
Antigen processing and presentation of endogenous peptide antigen via MHC class I (GO: 0019885)	0.03867	0.3282	*ERAP2*
Regulation of interleukin-6 production (GO: 0032675)	0.03905	0.3282	*IFNG*, *SPHK2*, *C5AR2*
Interferon-mediated signal pathway (GO: 0140888)	0.04099	0.3282	*IFNLR1*, *IFNG*
Term (GO: biological process) with downregulated genes	*p*-value	*q*-value	DEGs
Oxygen transport (GO: 0015671)	0.00061	0.04603	*HBB*, *HBA1*
Negative regulation of T cell proliferation (GO: 0042130)	0.00805	0.2279	*GNPMB*, *TNFRS21*
Positive regulation of receptor binding (GO: 1900122)	0.01861	0.2279	*MMP9*
T-helper 17 cell lineage commitment (GO: 0072540)	0.02596	0.2279	*SOCS3*
Lymphocyte apoptotic process (GO: 0070227)	0.02596	0.2279	*TNFRS21*
Neutrophil chemotaxis (GO: 0030593)	0.02844	0.2279	*CXCL10*, *KLF5*
T cell chemotaxis (GO: 0010818)	0.03688	0.2279	*CXCL10*
Cellular response to UV-A (GO: 0071492)	0.03688	0.2279	*MMP9*

**Table 7 biomedicines-13-02998-t007:** KEGG pathway terms for upregulated and downregulated genes in PBMCs from patients with pulmonary sarcoidosis with vs. those without extrapulmonary lesions. Relevant terms were excerpted.

Term (KEGG Pathway) with Upregulated Genes	*p*-Value	*q*-Value	DEGs
Herpes simplex virus 1 infection	0.00011	0.00908	*IFNG*, *TNFSF14*, *ZNF211*, *470*, *2*, *133*, *620*, *473*, *569*, *615*, *ZFP82*
Protein export	0.00984	0.4135	*IMMP1L*, *IMMP2L*
D-Glutamine and D-glutamate metabolism	0.03233	0.7322	*GLUD2*
Term (KEGG pathway) with downregulated genes	*p*-value	*q*-value	DEGs
TNF signaling pathway	0.008617	0.2019	*MMP9*, *CXCL10*, *SOCS3*
IL-17 signaling pathway	0.04861	0.4739	*MMP9*, *CXCL10*

## Data Availability

The data presented in this study are available upon request from the corresponding author. The datasets analyzed or generated during the study can be found in the GEO database. The name of the repository and accession number can be found below: https://www.ncbi.nlm.nih.gov/geo/query/acc.cgi, accession number GSE302854.

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
