# Peer review of "Differential Transcriptomic Features of Peripheral Blood Mononuclear Cells in Pulmonary Sarcoidosis with and Without Extrapulmonary Lesions in an East Asian Population"

_biomedicines, 2025, doi:10.3390/biomedicines13122998_

Round 1

Reviewer 1 Report

Comments and Suggestions for Authors

The authors have conducted an excellent study providing great insight to the differential transcriptomic features of PBMCs in patients with sarcoidosis vs. healthy controls. In addition, they have compared the transcriptomic features in PBMCs of sarcoidosis pts with extrapulmonary vs. isolated pulmonary disease. Their findings are in accordance with available literature and provide re-assurance regarding the current state of knowledge in sarcoidosis. 

I note however, that it is important to state clearly that one is not able to tell whether these findings are are consequence of the disease (I suspect this is the case) versus contribute or influence disease development. The authors have stated this clearly in the limitations section and I am happy with their statement to this effect. 

I would also however, that the authors make an effort to include more recent publications in their references. Majority of the references are from the early 2000s and some from the 1980s and 1990s. More recent work has been done in this field and should be references. In addition, the authors should also make sure that the references line up as stated. For example - Lines 388 - 390 speak to the diagnostic algorithm and list "ref 11". In looking at the references, "ref 11" is NOT the diagnostic algorithm references. This needs to be addressed. 

Reviewer 2 Report

Comments and Suggestions for Authors

In this paper, the authors describe peripheral blood mononuclear cells (PBMC) transcriptomic analysis using RNA sequencing in a cohort of sarcoidosis patients (N =15, distributed as 9 with extrapulmonary lesions (EPL) and 5 without) and compared the results to 26 healthy controls (HC).

The author processed the result by 1) principal component analysis (PCA), 2) enrichment analysis, and 3) Gene Ontology (GO) and KEGG pathway enrichment analyses.

The paper is interesting, well-executed, and relevant.

I have the following comments.

Abstract

I think the section will be easier to read if a more stringent division between introduction, method, results, and so on is used.

Importantly, no patient was on immunosuppression at the time of inclusion, which should be noted.

Introduction

Well written

Line 50-57. I agree that PBMCs are involved in the inflammation of sarcoidosis. They are probably not all equally involved, and the question is whether a different result would have been obtained if cell sorting had been done before analysis. Please comment in the discussion section.

Materials and Methods

Line 394. How did you define cardiac and ocular sarcoidosis? By Judson 2014 (The WASOG Sarcoidosis Organ Assessment Instrument: An update of a previous clinical tool), HRS criteria 2014, JCS criteria 2016?

Is reference 11 correct (line 391)?

Results

The group af EPL patients is very heterogeneous. At least for this reader, it seems somewhat speculative regarding whether patients with cutaneous sarcoidosis exhibit the same gene activation as those with ocular sarcoidosis. Please comment in the discussion section.

In Table 6 : A typo? ”cellular response to PH”     ….pH?

Discussion

Comprehensive

Line 235. Please add a reference.

The observations regarding cardiac sarcoidosis are speculative, as the authors also point out. Line 339. Consider reducing this subsection.

Reviewer 3 Report

Comments and Suggestions for Authors

The reviewed manuscript entitled “Differential Transcriptome Features of Peripheral Blood Mononuclear Cells in Pulmonary Sarcoidosis with and without Extrapulmonary Lesions” examine the gene expression profile of peripheral blood mononuclear cells of a cohort of people with or without pulmonary sarcoidosis. The authors further segregated the subjects with pulmonary sarcoidosis into those with or without extrapulmonary lesions. An RNA-seq analysis and subsequent bioinformatic pipeline suggested the expression of 227 genes to be altered during pulmonary sarcoidosis. A list of 206 genes is also associated with extrapulmonary sarcoidosis. Though the manuscript seems interesting, the lack of validation of the suggested candidate genes with specific techniques, e.g., qPCR, western blot, and multiplex immunoassay, renders the integrity of the study findings questionable. If possible, verification of 2-3 candidates could be included in the present work.

Specific comments

  • Refer to the ethnicity of the studied subjects in the manuscript title.
  • Numbers of the up- and downregulated genes for the two conducted RNA-seq analyses should be written in the abstract. The top 10 up- and downregulated genes should be listed in the abstract and the manuscript context as well.
  • Demographic data of the study subjects should be moved to the “Material and methods” section.
  • It is standard in bioinformatic studies to use FDR-adjusted p-value (or q-value) instead of p-value. It is not clear whether this approach was used or not in this study.
  • If possible, names of top up- and downregulated transcripts should be named on volcano plots of Figures 1 and 2.
  • The addition of a number of network analyses using web-based software applications, e.g., Ingenuity Pathway Analysis (IPA), will help for better visualization of the study findings and outcomes.
  • Line 226: What do you mean by “Table 2a”?
  • The discussion should focus on the novel findings and their functional interpretations. It is not recommended to cite tables and figures in this section.

Reviewer 4 Report

Comments and Suggestions for Authors

biomedicines-3979943 Differential Transcriptome Features of Peripheral Blood Mononuclear Cells in Pulmonary Sarcoidosis with and without Extrapulmonary Lesions

In this study, authors evaluated a cohort of patients with sarcoidosis and healthy controls, of similar age and sex distribution to evaluate how PBMC gene expression is associated with sarcoidosis lesion progression. Also, gene expression patterns associated with the development of extrapulmonary lesions (EPL) in pulmonary sarcoidosis were investigated.

Overall, the manuscript suffers from some weaknesses. Therefore, it is not recommended for publication at its current state. My comments in detail are as following:

 Abstract

  • The abstract should be written in an informative style. Please give real values/data, not vague subjective terms and avoid generalizations and nonessential information in the results.

Introduction

  • Page 2, Line 43: “Pulmonary sarcoidosis is the most frequent manifestation of the disease, with pulmonary involvement observed in 86% of Asian and 95% of Caucasian and African American patients [3, 4].” Cited references are very old. More current references may be used.
  • Page 2, Line 51-57: References should be added.

Discussion

  • In the current study, the authors reported analyzing a cohort of patients with sarcoidosis and healthy control with similar age and sex distribution to assess how PBMC gene expression correlates with sarcoidosis lesion progression. What new results were obtained in this study compared to their previous study? A commentary paragraph comparing these findings should be added. It should also be explained how this study contributes to the findings of the previous study.
  • Page 12 Line 231-234: References should be added.

Materials and methods

  • The study population (number of sarcoidosis and healthy controls) is not presented in the “Subject” section.
  • "This study included 14 patients with pulmonary sarcoidosis and 26 HCs". Although no significant differences in age or sex were observed between the pulmonary sarcoidosis and HC groups. the numbers of individuals in the compared populations sufficient? Was a statistical analysis performed prior to show equality of variance between groups?

Round 2

Reviewer 3 Report

Comments and Suggestions for Authors

The authors responded to my comments and the manuscript is improved. The manuscript can be considered for publication in its present form. Just a minor notice, please include full description of gene symbols listed on Figures 1B and 2B to the relevant parts of the figure legend.

Reviewer 4 Report

Comments and Suggestions for Authors

- I am pleased to inform you that the authors have addressed all the reviewers’ comments and have considerably revised their manuscript. So, I recommend accepting this manuscript.
